# Associations of Serum Magnesium with Brain Morphology and Subclinical Cerebrovascular Disease: The Atherosclerosis Risk in Communities-Neurocognitive Study

**DOI:** 10.3390/nu13124496

**Published:** 2021-12-16

**Authors:** Aniqa B. Alam, DaNashia S. Thomas, Pamela L. Lutsey, Srishti Shrestha, Alvaro Alonso

**Affiliations:** 1Department of Epidemiology, Emory University School of Public Health, Atlanta, GA 30322, USA; alvaro.alonso@emory.edu; 2Department of Psychology, School of Arts and Sciences, Clark Atlanta University, Atlanta, GA 30314, USA; danashiathomas@gmail.com; 3Division of Epidemiology and Community Health, University of Minnesota School of Public Health, Minneapolis, MN 55454, USA; lutsey@umn.edu; 4Department of Neurology and the MIND Center, University of Mississippi Medical Center, Jackson, MS 39216, USA; sshrestha1@umc.edu

**Keywords:** magnesium, brain volume, cerebrovascular disease

## Abstract

Circulating magnesium has been associated with a lower risk of dementia, but the physiologic effects by which magnesium may prevent neurological insults remain unclear. We studied 1466 individuals (mean age 76.2 ± 5.3, 28.8% black, 60.1% female) free of prevalent stroke, with measured serum magnesium and with available MRI scans obtained in 2011–2013, participating in the Atherosclerosis Risk in Communities Neurocognitive Study (ARIC-NCS). Cross-sectional differences in frontal, temporal, parietal, and occipital lobe volume, along with deep grey matter, total brain, and white matter hyperintensity volume across serum magnesium (categorized into quintiles and per standard deviation increases) were assessed using multiple linear regression. We also examined associations of magnesium with the prevalence of cortical, subcortical, and lacunar infarcts using multiple logistic regression. After adjusting for demographics, biomarkers, medications, and cardiometabolic risk factors, higher circulating magnesium was associated with greater total brain volume and frontal, temporal, and parietal lobe volumes (volumes 0.14 to 0.19 standard deviations higher comparing Q5 to Q1). Elevated magnesium was also associated with lower odds of subcortical infarcts (OR (95%CI): 0.44 (0.25, 0.77) comparing Q5 to Q1) and lacunar infarcts (OR (95%CI): 0.40 (0.22, 0.71) comparing Q5 to Q1). Elevated serum magnesium was cross-sectionally associated with greater brain volumes and lower odds of subclinical cerebrovascular disease, suggesting beneficial effects on pathways related to neurodegeneration and cerebrovascular damage. Further exploration through prospective analyses is needed to assess increasing circulating magnesium as a potential neuroprotective intervention.

## 1. Introduction

Magnesium serves multiple functions in the body, including involvement in cognition. The mineral has been associated with lower risk of dementia in community-based studies [1,2] and has shown neuroprotective effects in animal models of dementia [3,4], but the exact mechanisms through which magnesium may prevent neurological insults remain unclear. 

Magnesium deficiency has been linked with increased risk of hypertension [5], cardiovascular diseases [6], and thrombosis [7], which are established risk factors for dementia [8]. The cutoff for hypomagnesemia varies across the literature, but typically ranges from 0.60 mmol/L to 0.66 mmol/L [9,10]. Magnesium promotes the synthesis of nitric oxide, which itself is protective against thrombosis and hypertension due to its anti-platelet properties and ability to induce vasodilation [11]. Magnesium is also a natural antagonist to calcium, which has been known to encourage the over-excitation–and subsequent death–of neurons [12], making it important within the context of dementia. Elevated serum magnesium is associated with a lower risk of cardioembolic stroke [13] and a lower risk of death in patients with acute ischemic stroke [14]. Conversely, lower magnesium can predict ischemic stroke events and the need for carotid revascularization in patients with severe atherosclerosis [15]. 

Evaluating the association of circulating magnesium with brain imaging markers of neurodegeneration and cerebrovascular disease may offer a clearer picture on the underlying mechanisms linking magnesium and dementia. Thus, we analyzed data from 1466 participants in the Atherosclerosis Risk in Communities (ARIC)-Neurocognitive Study (ARIC-NCS) with available data on circulating magnesium and brain imaging to determine the association of magnesium with brain volumes, infarcts, and white matter disease.

## 2. Materials and Methods

### 2.1. Study Population

The ARIC study is an ongoing prospective cohort of 15,792 participants from 4 communities across the US: Jackson, Mississippi; Washington County, Maryland; Forsyth County, North Carolina; and selected suburbs of Minneapolis, Minnesota. The baseline visit took place from 1987 to 1989. Details of the ARIC cohort have been published elsewhere [16]. As part of ARIC-NCS, an ancillary study to ARIC in visit 5 (2011–2013), participants were invited to undergo in-person neurocognitive examinations, administered by a trained nurse. Those who were found to have cognitive impairments, as well as a subset of cognitively normal participants, were invited to undergo further assessment and a brain MRI [17]. Participants at each visit provided written informed consent. The institutional reviews boards at each participating center approved of the ARIC study protocol. The Institutional Review Board of Emory University approved the present study (IRB00088867, initial approval date 2 August 2016). 

From 6538 participants in ARIC visit 5, we included those with available serum magnesium measurements and participation in the MRI study (*n* = 1577). Those with prevalent stroke were excluded from analysis (*n* = 101). Due to low counts, Asian and Native American participants (*n* = 5) and black participants from Washington County (*n* = 5) were excluded. After applying our inclusion and exclusion criteria, 1466 participants were included in the analysis (Figure 1). 

### 2.2. Brain Imaging

Imaging protocols have been described at length elsewhere [18]. Briefly, brain volumes were estimated from T1-weighted MP-RAGE sequences using the FreeSurfer system (FreeSurfer, http://surfer.nmr.mgh.harvard.edu, last accessed 9 December 2021) [19]. Infarcts and white matter hyperintensity (WMH) volumes were derived from T2-weighted fluid attenuation inversion recovery (FLAIR) sequences. Lobe volumes of interest included the frontal, temporal, parietal and occipital lobes, along with deep grey matter and total brain volume. Cerebrovascular disease was characterized by the presence of cortical or subcortical infarcts; subcortical infarcts were further classified as lacunar infarcts if they measured ≤20 mm and were located in the caudate, lenticular nucleus, internal capsule, thalamus, brainstem, deep cerebellar white matter, centrum semiovale, or corona radiate [20].

### 2.3. Serum Magnesium

Blood was drawn into vacuum tubes, stored at −80 °C and later sent to the ARIC central laboratories. Serum magnesium was measured using the xylidyl blue-I method with a Roche COBAS 6000 chemistry analyzer (Roche Diagnostics, Indianapolis, IN, USA) [21]. Measurements from 242 duplicate samples returned a coefficient of variation of 1.9%.

### 2.4. Covariates

All covariates of interest were measured at visit 5, except for education, which was recorded at baseline. Education was defined as either “less than high school” or “high school and above”. Sodium, potassium, calcium, estimated glomerular filtration rate, c-reactive protein and HDL and total cholesterol were measured in fasting blood samples taken at visit 5. We accounted for sodium and potassium as these minerals work alongside magnesium and calcium to regulate NMDA receptor functioning [22,23]. Current smoking status was determined through self-report. Body mass index (BMI) was measured during the in-person assessments for visit 5. Use of hypertension medication was determined at the in-person assessments where participants were asked to bring in their prescription medication. Blood pressure was measured three times in person and the mean of the second and third measurements was used to define hypertension (systolic blood pressure ≥140 mmHg or diastolic blood pressure ≥90 mmHg or currently taking antihypertensive medication). Diabetes status was based on having a fasting glucose of 126 mg/dL or greater, a non-fasting glucose of 200 mg/dL or greater, using medications for diabetes or self-report of diabetes by a physician. Prevalent coronary heart disease (CHD) was defined as having been hospitalized for myocardial infarction (MI), having an MI as determined by ECG, fatal CHD or having a cardiac procedure prior to visit 5. Prevalent heart failure (HF) was defined using self-reported information collected at the baseline visit and from adjudicated events during follow-up. Participants also provided consent for the genotyping of the APOE gene. 

### 2.5. Statistical Analysis

We used multiple linear regression to examine the association of serum magnesium (in approximate quintiles and 1-standard deviation increases) with brain volumes, including the frontal, temporal, parietal and occipital lobes, along with deep grey matter and total brain volume. Each brain volume was scaled using its standard deviation to make easier comparison of associations across different brain regions. We used logistic regression to examine associations of magnesium with markers of subclinical cerebrovascular disease, characterized by the presence of cortical, subcortical, and lacunar infarcts. We also assessed the association of magnesium with log-transformed WMH volumes.

Model 1 adjusted for age at visit 5, race-center (Forsyth/White, Forsyth/Black, Minneapolis/White, Washington/White, Jackson/Black), sex, education and, for the analyses of brain and WMH volumes, estimated total intracranial volume. Model 2 was further adjusted for calcium, sodium, potassium, BMI, HDL and LDL cholesterol, smoking status, hypertension, hypertension medication use, estimated glomerular filtration rate, c-reactive protein, diabetes, prevalent CHD, and prevalent HF, along with the presence of the *APOE* ɛ4 allele. We applied sampling weights that were standardized to visit 5 attendance and also account for selection for brain MRI. 

## 3. Results

The 1466 participants included in this analysis had a mean age of 76.2 years (SD: 5.3), were 28.8% black and 60.1% female. Those with higher serum magnesium were more educated, more likely to be white and less likely to suffer from hypertension and diabetes (Table 1). It is also worth noting that most participants included are above normal BMI (BMI ≥ 25).

Elevated serum magnesium was associated with greater total brain volume and greater volumes for most lobes at visit 5, with some evidence of a linear association (Table 2), in both the minimally and fully adjusted models. After adjustment for model 2 covariates, 1-standard deviation (0.08 mmol/L) higher serum magnesium was associated with 0.03 (occipital lobe volume) to 0.06 (total brain and parietal lobe volumes) higher volumes (in standard deviation units) (Appendix A). Further examination of sex and race interactions with magnesium found no consistent differences in these associations of magnesium with brain volumes across these groups, except in the parietal lobe for sex (*p* = 0.03, stronger association in males than females) (Appendix A) and in the frontal lobe for race (*p* = 0.02, stronger association in Black participants) (Appendix A).

Elevated magnesium was associated with lower odds of subcortical infarcts (OR: 0.44, 95%CI: 0.25, 0.77 comparing Q5 to Q1, OR: 0.77, 95%CI: 0.65, 0.91 per 1-standard deviation increase) and lacunar infarcts (OR: 0.40, 95%CI: 0.22, 0.71 comparing Q5 to Q1; OR: 0.76, 95%CI: 0.64, 0.89 per 1-standard deviation increase) (Table 3). In contrast, circulating magnesium was not associated with cortical infarctions and WMH volumes. Furthermore, no differences were detected across sex and race, as evidenced by nonsignificant interaction terms (Appendix A).

## 4. Discussion

Within this community-based cohort, we found cross-sectional associations of elevated serum magnesium with greater total brain volume and the volume of most specific brain lobes. Furthermore, elevated serum magnesium was also associated with lower odds of subcortical and lacunar infarcts.

Lower circulating magnesium has been associated with higher risk of dementia [1] and stroke [13], but the mechanisms are not exactly known. Memory formation and other learning processes are informed by n-methyl-d-aspartate receptor activity [24], the over-excitation of which can lead to cell death [12]. Magnesium can inhibit these receptors and prevent neurodegeneration [25], which may explain the larger brain volumes in those with the highest levels of magnesium. Moreover, myelin integrity is impaired in dementia [26]. Magnesium deficient rats were found to have thinner myelin sheaths and fewer myelinated axons overall compared to non-deficient rats [27]. In addition, magnesium sulfate has shown to uphold the integrity of the blood brain barrier by preventing the production of cytokines and other markers of oxidative stress [28]. Furthermore, because magnesium is a natural antagonist to calcium and its inflammatory properties [29], higher circulating magnesium may reduce the risk of ischemic stroke [30], and improve endothelial function [31]. These mechanisms, overall, may explain the associations between circulating magnesium and brain volumes in the ARIC cohort.

That said, of all the magnesium in the body, only 0.3% is in the blood [32]. Magnesium in the body can be primarily found in the bones, soft tissues, and teeth [33]. Though serum magnesium only represents a small portion of the total body magnesium, serum measurements may still have some utility in predicting brain size and health, particularly when considering blood brain barrier (BBB) permeability. The tight junctions of the BBB can be an issue in the transport of micronutrients into the cerebrospinal fluid (CSF) and brain [34]. Yet, low molecular weight magnesium actually has a better chance of passing through this barrier than high molecular weight metals such as iron, copper, and zinc [35], which are involved in the accumulation of amyloid plaques in the brain [36]. In instances of traumatic brain injury, studies have reported decreases in both brain magnesium [37] and serum magnesium [38]. Thus, while the extent to which serum magnesium is correlated with central nervous system magnesium is unknown, there appears to be some degree of positive association. 

We did not find a clear relationship of circulating magnesium with WMH volume and the presence of cortical infarcts, but this is also consistent with previous studies. For example, administering magnesium sulfate to hypertensive rats did not appear to affect the infarct volume; interestingly enough, it did appear to attenuate the motor impairments that would accompany such white matter damage [39]. That said, we did find elevated magnesium to have a protective effect against subcortical infarcts. The IMAGES randomized clinical trial yielded similar results in that while intravenous magnesium sulfate given in the setting of acute stroke did not appear to affect outcomes for disability and death globally, there was a significant reduction in poor outcomes for those with non-cortical stroke, particularly those with lacunar clinical syndromes [40]. 

There are several strengths to our study. To our knowledge, this is the first study to look at the relationship of serum magnesium with brain volumes and subclinical indicators of cerebrovascular disease. Furthermore, we account for a wide array of potential confounders including anthropomorphic measurements, lifestyle factors, biomarkers, medication use, and clinical CVD risk factors. That said, some limitations present in the study warrant a cautious interpretation of our results. First, the study is cross-sectional in nature with one-time measurements of serum magnesium, brain volumes, and brain lesions, making it difficult to parse the directionality of the associations. Second, many participants had died by visit 5 or were alive but refused or were unwilling to attend visits. Thus, participants selected into the study may have suffered less neurodegeneration and damage than those lost to follow-up, which could lead to potential selection bias. Third, despite adjustment for a wide array of covariates, there may be additional factors not accounted for in our models.

## 5. Conclusions

We found elevated serum magnesium to be associated with greater brain volumes and lower odds of subclinical cerebrovascular disease compared to those with low circulating magnesium, potentially implicating protection against neuronal degeneration and cerebrovascular disease as mechanisms responsible for the lower risk of dementia in those with higher circulating magnesium. These findings should be confirmed in well-designed prospective analyses, with particular focus on evaluating the potential of interventions aimed at increasing circulating magnesium for the prevention of neurodegeneration and cerebrovascular disease. Given the established effect of oral magnesium supplementation on circulating magnesium concentrations [41], these interventions have the potential to make a significant impact on the prevention of dementia, a major contributor to the burden of disease in the population.

## Figures and Tables

**Figure 1 nutrients-13-04496-f001:**
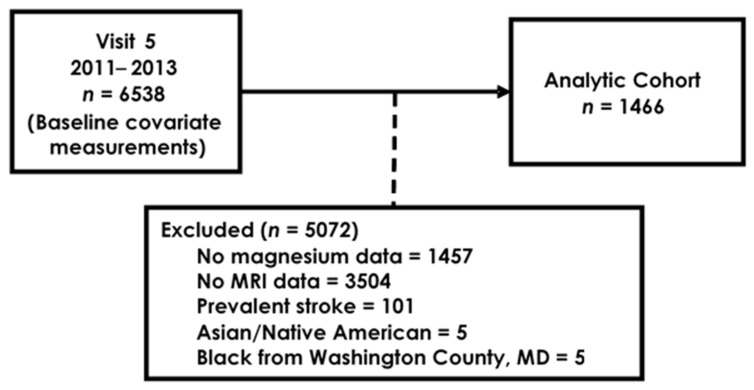
Flow diagram of ARIC participant selection into study.

**Table 1 nutrients-13-04496-t001:** Participant characteristics by magnesium quintiles, ARIC-NCS 2011–2013.

	Q1	Q2	Q3	Q4	Q5
Magnesium, mmol/L	<0.76	0.76–0.80	0.81–0.84	0.85–0.88	>0.88
N	277	226	296	351	316
Age, years	75.3 (5.1)	76.9 (5.1)	76.0 (5.5)	76.1 (5.0)	76.7 (5.5)
Female, %	71.1	58.4	56.1	55.3	60.8
Black, %	39.0	36.3	28.7	23.9	19.9
More than high school, %	38.3	42.9	44.3	45.3	46.5
HDL cholesterol, mg/dL	51.3 (14.7)	52.8 (13.4)	53.6 (14.1)	52.8 (13.7)	53.5 (14.5)
Total cholesterol, mg/dL	174.9 (42.0)	176.0 (38.5)	184.3 (42.5)	184.3 (42.4)	188.2 (41.7)
BMI, kg/m^2^	30.3 (6.3)	29.1 (6.0)	28.2 (5.6)	27.6 (4.8)	27.3 (5.0)
Sodium, mmol/L	138.3 (3.2)	138.9 (2.8)	139.1 (2.6)	139.1 (2.4)	139.6 (2.5)
Potassium, mmol/L	4.0 (0.4)	4.0 (0.3)	4.0 (0.3)	4.1 (0.3)	4.1 (0.4)
Calcium, mg/dL	9.5 (0.4)	9.4 (0.3)	9.3 (0.4)	9.4 (0.4)	9.4 (0.4)
Current smoking, %	7.2	5.8	4.7	4.3	4.1
Hypertension, %	87.7	82.7	72.3	67.0	63.9
Hypertension medication, %	81.2	77.4	63.5	59.0	55.1
Diabetes, %	56.0	34.5	26.7	27.4	19.9
Coronary heart disease, %	7.6	12.0	8.1	8.0	10.1
Heart failure, %	9.8	14.2	9.8	6.3	8.9
eGFR, mL/min/1.73m^2^	66.1 (19.3)	66.0 (17.4)	67.8 (17.7)	67.4 (16.5)	62.9 (17.7)
C-reactive protein, mg/L	4.7 (8.1)	3.9 (6.7)	3.8 (5.4)	3.4 (4.9)	3.4 (6.5)
APOE ɛ4 allele, %	26.4	31.0	30.1	29.6	28.5
Markers of subclinical cerebrovascular disease					
Cortical infarcts, %	9.8	10.2	8.8	9.4	7.3
Subcortical infarcts, %	21.7	20.4	15.2	16.2	13.6
Lacunar infarcts, %	21.3	19.5	15.2	16.0	12.7
White matter hyperintensity volume (cm^3^), (SD)	17 (17)	19 (18)	16 (16)	17 (16)	19 (20)
Brain volume (cm^3^), (SD)					
Total brain	983 (101)	1007 (106)	1024 (107)	1020 (99)	1019 (118)
Frontal	145 (15)	149 (16)	153 (15)	151 (14)	151 (17)
Temporal	99 (10)	101 (12)	103 (11)	103 (11)	102 (13)
Occipital	39 (5)	40 (6)	41 (5)	41 (5)	41 (6)
Parietal	102 (12)	105 (12)	107 (12)	107 (12)	107 (14)
Deep grey matter	42 (4)	42 (4)	43 (4)	43 (4)	43 (5)

Values correspond to means (SD) or percentage.

**Table 2 nutrients-13-04496-t002:** Associations of serum magnesium with brain volumes, ARIC-NCS 2011–2013.

	**Q1**	**Q2**	**Q3**	**Q4**	**Q5**	**1-SD Mg**
Total brain volume						
Model 1 *	Ref.	0.11 (0.02, 0.20)	0.19 (0.10, 0.28)	0.15 (0.06, 0.23)	0.21 (0.12, 0.30)	0.06 (0.04, 0.09)
Model 2 **	Ref.	0.09 (0.00, 0.18)	0.17 (0.08, 0.26)	0.13 (0.05, 0.22)	0.19 (0.10, 0.28)	0.06 (0.03, 0.09)
Frontal lobe						
Model 1 *	Ref.	0.08 (−0.04, 0.19)	0.21 (0.11, 0.32)	0.11 (0.01, 0.21)	0.19 (0.08, 0.29)	0.06 (0.02, 0.09)
Model 2 **	Ref.	0.07 (−0.05, 0.18)	0.19 (0.08, 0.30)	0.09 (−0.02, 0.19)	0.16 (0.05, 0.27)	0.05 (0.01, 0.08)
Temporal lobe						
Model 1 *	Ref.	0.10 (0.00, 0.23)	0.10 (−0.01, 0.20)	0.10 (0.04, 0.20)	0.15 (0.05, 0.25)	0.06 (0.02, 0.09)
Model 2 **	Ref.	0.11 (0.00, 0.22)	0.09 (−0.02, 0.20)	0.08 (−0.02, 0.19)	0.14 (0.03, 0.24)	0.05 (0.02, 0.09)
Occipital lobe						
Model 1 *	Ref.	0.14 (0.01, 0.28)	0.15 (0.02, 0.28)	0.15 (0.03, 0.27)	0.14 (0.02, 0.26)	0.05 (0.01, 0.09)
Model 2 **	Ref.	0.12 (−0.02, 0.26)	0.10 (−0.03, 0.23)	0.11 (−0.02, 0.23)	0.08 (−0.05, 0.21)	0.03 (−0.01, 0.08)
Parietal lobe						
Model 1 *	Ref.	0.14 (0.03, 0.25)	0.16 (0.05, 0.27)	0.13 (0.03, 0.23)	0.18 (0.08, 0.29)	0.06 (0.03, 0.10)
Model 2 **	Ref.	0.13 (0.02, 0.25)	0.14 (0.04, 0.25)	0.12 (0.02, 0.23)	0.17 (0.06, 0.28)	0.06 (0.03, 0.10)
Deep grey matter						
Model 1 *	Ref.	0.08 (−0.05, 0.21)	0.09 (−0.02, 0.21)	0.12 (0.00, 0.23)	0.14 (0.01, 0.27)	0.05 (0.00, 0.09)
Model 2 **	Ref.	0.06 (−0.07, 0.19)	0.08 (−0.04, 0.20)	0.11 (−0.01, 0.23)	0.13 (0.00, 0.26)	0.05 (0.00, 0.09)

Brain volumes modeled in standard deviation units. * Model 1 results from multiple linear regression adjusted for age, sex, race/center, education and total intracranial volume. 1-SD Mg: 0.08 mmol/L. ** Model 2 results from multiple linear regression adjusted for model 1, plus LDL and HDL cholesterol, body-mass index, sodium, potassium, calcium, smoking status, hypertension, hypertension medication use, history of coronary heart disease and heart failure, diabetes, eGFR, c-reactive protein, APOE allele. 1-SD Mg: 0.08 mmol/L.

**Table 3 nutrients-13-04496-t003:** Associations of serum magnesium with markers of cerebrovascular disease, ARIC-NCS 2011–2013.

Variable	Q1	Q2	Q3	Q4	Q5	1-SD Mg
	Odds Ratios (95%CI)					
Cortical infarcts						
Model 1 *	1 (ref.)	1.19 (0.60, 2.35)	1.13 (0.59, 2.18)	1.03 (0.55, 1.92)	0.84 (0.43, 1.65)	1.02 (0.84, 1.24)
Model 2 **	1 (ref.)	1.24 (0.61, 2.53)	1.25 (0.63, 2.49)	1.22 (0.62, 2.41)	1.01 (0.49, 2.07)	1.11 (0.90, 1.37)
Subcortical infarcts						
Model 1 *	1 (ref.)	0.89 (0.52, 1.51)	0.47 (0.28, 0.78)	0.56 (0.34, 0.92)	0.42 (0.25, 0.70)	0.75 (0.65, 0.88)
Model 2 **	1 (ref.)	0.97 (0.56, 1.68)	0.52 (0.30, 0.90)	0.58 (0.34, 0.99)	0.44 (0.25, 0.77)	0.77 (0.65, 0.91)
Lacunar infarcts						
Model 1 *	1 (ref.)	0.86 (0.51, 1.48)	0.47 (0.28, 0.78)	0.55 (0.33, 0.91)	0.38 (0.22, 0.65)	0.74 (0.64, 0.86)
Model 2 **	1 (ref.)	0.94 (0.54, 1.64)	0.52 (0.30, 0.90)	0.57 (0.34, 0.98)	0.40 (0.22, 0.71)	0.76 (0.64, 0.89)
	Beta (95%CI)					
Ln(WMH volume) ^†^						
Model 1 *	Ref.	−0.01 (−0.13, 0.16)	−0.15 (−0.32, −0.01)	−0.16 (−0.32, 0.00)	−0.12 (−0.26, 0.02)	−0.05 (−0.10, −0.01)
Model 2 **	Ref.	0.03 (−0.13, 0.18)	−0.10 (−0.25, 0.04)	−0.11 (−0.27, 0.05)	−0.07 (−0.22, 0.08)	−0.03 (−0.08, 0.02)

Results from logistic regression (infarcts) and linear regression (WMH volume) adjusted for: * Model 1: age, sex, race/center, and education. 1-SD Mg: 0.08 mmol/L. ** Model 2: model 1, plus LDL and HDL cholesterol, body-mass index, sodium, potassium, calcium, smoking status, hypertension, hypertension medication use, history of coronary heart disease and heart failure, diabetes, eGFR, c-reactive protein, APOE allele. 1-SD Mg: 0.08 mmol/L. ^†^ Additionally adjusted for total intracranial volume. WMH: white matter hyperintensities.

## Data Availability

The data presented in this study may be available on request from the corresponding author or through ARIC directly: https://sites.cscc.unc.edu/aric/distribution-agreements (last accessed 9 December 2021).

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
