# Peer review of "Associations of Serum Magnesium with Brain Morphology and Subclinical Cerebrovascular Disease: The Atherosclerosis Risk in Communities-Neurocognitive Study"

_nutrients, 2021, doi:10.3390/nu13124496_

Round 1

Reviewer 1 Report

The manuscript is well presented and clear.

Please mention the reference range of magnesemia in the introduction so that the readers can easily interpret your tables. Also please mention in the results that most of the people enrolled have a BMI higher than normal. 

Author Response

Response to the Reviewers

We thank the reviewers for their feedback. Below, we provide a point-by-point response to their comments and describe changes made to the manuscript.

Reviewer 1:

The manuscript is well presented and clear.

Please mention the reference range of magnesemia in the introduction so that the readers can easily interpret your tables. Also please mention in the results that most of the people enrolled have a BMI higher than normal. 

  • We have included a statement in the introduction mentioning the cutoff for hypomagnesemia (lines 42-43) and a brief observation on the BMI of the participants (line 139).

Reviewer 2 Report

The paper describes interesting associations between serum Mg concentration and brain parameters in the ARIC study. Aiming to publish the results in a Nutrition Journal the authors should focus also on the aspects of Mg in serum being influenced by dietary Mg.

Generally, the authors do not comment on serum Mg regarding being adequate or hypomagnesemic. The range of serum Mg in the 5 quintiles should be given (probably it would be preferable to use mmol/l units then mg/dl), for example in Q2 is it from 1.81 to 1.95 mg/dl and then Q3 1.96 to 2.05 mg/dl?

Generally, the presentation of data is complicated to understand. The authors assume a linear correlation of brain parameters with serum Mg concentration – what if there is a threshold at about 0.8 or 0.85 mmol/l as discussed by others (https://doi.org/10.3945/an.116.012765)? The presentation of the brain parameters as plain numbers or odd ratios per quartile (in a graph) would make the understanding much easier.

There is a large difference regarding hypertension and diabetes incidence in the quartiles. Both is significantly connected to serum Mg. Is it really not that much influencing the outcome? Can you get rid of that by Model 2? Is the number of cases then high enough to get statistically significant outcomes? The authors did not comment on the major confounding parameters in the discussion.

What exactly is the statistical significance of the results if one looks at table 1 it does not seem to be very impressive regarding volume parameters. The authors should discuss more in detail.

There exists a difference between serum Mg and Mg in the cerebrospinal fluid and Mg does not equilibrate easily across the blood brain barrier. Authors should comment.

The relevance for prevention must be explained. 

Title of Tab 3 is wrong

Author Response

We thank the reviewer for their feedback. Below, we provide a point-by-point response to their comments and describe changes made to the manuscript.

Reviewer 2:

The paper describes interesting associations between serum Mg concentration and brain parameters in the ARIC study. Aiming to publish the results in a Nutrition Journal the authors should focus also on the aspects of Mg in serum being influenced by dietary Mg.

 Generally, the authors do not comment on serum Mg regarding being adequate or hypomagnesemic. The range of serum Mg in the 5 quintiles should be given (probably it would be preferable to use mmol/l units then mg/dl), for example in Q2 is it from 1.81 to 1.95 mg/dl and then Q3 1.96 to 2.05 mg/dl?

  • Unfortunately, due to the limitations in the precision of the instrument, serum Mg measurements are no more precise than those provided in the paper (i.e.: 1.9 mg/dL as opposed to 1.92 or 1.87 mg/dL). We have however changed the Mg units from mg/dL to mmol/L as suggested. Also, we provide a range of the concentrations that would be included in each category (e.g. in the 1.9 mg/dL, actual concentrations would range from 1.85 to 1.94 mg/dL, which corresponds to 0.76 to 0.80 mmol/L)

Generally, the presentation of data is complicated to understand. The authors assume a linear correlation of brain parameters with serum Mg concentration – what if there is a threshold at about 0.8 or 0.85 mmol/l as discussed by others (https://doi.org/10.3945/an.116.012765)? The presentation of the brain parameters as plain numbers or odd ratios per quartile (in a graph) would make the understanding much easier.

  • By categorizing serum Mg measurements into quintiles, we actually assume a non-linear association and are able to study thresholds. The hypothesis here is that those with the highest levels of Mg are in some way different than those with the lowest levels. Previous work support this rationale and method (DOI: 3390/nu12103074, 10.3389/fnagi.2020.00101). The overall interpretation of the findings suggests that there is some linear association with brain volumes and infarcts, as suggested by modeling Mg as a continuous variable.
  • We prefer to keep the presentation of results as is. The associations with brain volumes are presented using standardized measures to facilitate comparison across the different regions. Presentation of the differences in brain volumes in plain, raw numbers would not particularly help in the interpretation of the results. For table 3, presenting odds ratios and 95% confidence intervals of the association of serum Mg quintiles with prevalence of different types of infarcts allows us to evaluate the dose-response association.

There is a large difference regarding hypertension and diabetes incidence in the quartiles. Both is significantly connected to serum Mg. Is it really not that much influencing the outcome? Can you get rid of that by Model 2? Is the number of cases then high enough to get statistically significant outcomes? The authors did not comment on the major confounding parameters in the discussion.

  • We agree that hypertension and diabetes are significantly associated with serum Mg and would influence our outcome; that is why we account for hypertension and diabetes – and subsequent potential confounding due to these conditions – in our second model. This is a standard approach in observational studies. To exclude all those with hypertension and diabetes within the cohort would mean losing nearly 80% (N = 1,160) of the participants. Furthermore, it would not be a representative sample of this older, community-based cohort, and generalizability of the results would be compromised. We recognize, however, the potential for uncontrolled confounding due to hypertension and diabetes, and recognize this limitation in lines 224-227.

What exactly is the statistical significance of the results if one looks at table 1 it does not seem to be very impressive regarding volume parameters. The authors should discuss more in detail.

  • The goal of Table 1 is to describe the study sample by categories of the primary exposure, and not necessarily to draw conclusions on statistical significance. Of note, even if the brain volumes presented in Table 1 do not seem to vary widely across serum Mg quintiles (e.g. the crude difference between quintile 5 and quintile 1 for total brain volume is just 36 cm3), those differences correspond to sizable differences (approximately 0.3-0.4 standard deviations). Tables 2 and 3 are more appropriate for drawing conclusions on the plausibility of our hypothesis, since they adjust for all potential confounders.

There exists a difference between serum Mg and Mg in the cerebrospinal fluid and Mg does not equilibrate easily across the blood brain barrier. Authors should comment.

  • This is an important point. We have added a brief portion in the discussion section addressing the potential dynamic between serum Mg and CNS Mg:

“That said, of all the magnesium in the body, only 0.3% is in the blood.[30] The magnesium in the body can be primarily found in the bones, soft tissues, and teeth.[31] Though serum magnesium only represents a small portion of the total body magnesium, serum measurements may still have some utility in predicting brain size and health, particularly when considering blood brain barrier (BBB) permeability. The tight junctions of the BBB can be an issue in the transport of micronutrients into the cerebrospinal fluid (CSF) and brain.[32] Yet, low molecular weight magnesium actually has a better chance of passing through this barrier than high molecular weight metals such as iron, copper, and zinc,[33] which are involved in the accumulation of amyloid plaques in the brain.[34] In instances of traumatic brain injury, studies have reported decreases in both brain magnesium[35] and serum magnesium[36]. Thus, while the extent to which serum magnesium is correlated with central nervous system magnesium is unknown, there appears to be some degree of positive association.” (lines 194-206)

The relevance for prevention must be explained. 

  • In the second paragraph of the discussion we have laid out possible mechanisms and roles of magnesium in neurodegeneration, dementia, and overall memory formation and cognitive decline (lines 180-193). In conjunction with previous reports of elevated serum magnesium being associated with greater cognitive functioning compared to those at the lower end of the spectrum (also briefly discussed in lines 35-39 in the introduction), we believe this to be sufficient explanation of the potential benefits of increasing circulating magnesium in the body.

Title of Tab 3 is wrong

  • Thank you for catching this. The title has been corrected.

Round 2

Reviewer 2 Report

Thanks for making the paper clearer now.